# The Differences in Postprandial Serum Concentrations of Peptides That Regulate Satiety/Hunger and Metabolism after Various Meal Intake, in Men with Normal vs. Excessive BMI

**DOI:** 10.3390/nu11030493

**Published:** 2019-02-26

**Authors:** Edyta Adamska-Patruno, Lucyna Ostrowska, Joanna Goscik, Joanna Fiedorczuk, Monika Moroz, Adam Kretowski, Maria Gorska

**Affiliations:** 1Clinical Research Centre, Medical University of Bialystok, MC Sklodowskiej 24A, 15-276 Bialystok, Poland; joanna.goscik@umb.edu.pl (J.G.); j.fiedorczuk@wp.pl (J.F.); monika_bakun@wp.pl (M.M.); adamkretowski@wp.pl (A.K.); 2Department of Dietetics and Clinical Nutrition, Medical University of Bialystok, Mieszka I-go 4B, 15-054 Bialystok, Poland; lucyna.ostrowska@umb.edu.pl; 3Department of Endocrinology, Diabetology and Internal Medicine, Medical University of Bialystok, MC Sklodowskiej 24A, 15-276 Bialystok, Poland; mgorska@wp.pl

**Keywords:** obesity, postprandial adiponectin, postprandial leptin, postprandial total ghrelin, high-carbohydrate meal, high-fat meal

## Abstract

The energy balance regulation may differ in lean and obese people. The purposes of our study were to evaluate the hormonal response to meals with varying macronutrient content, and the differences depending on body weight. Methods. The crossover study included 46 men, 21–58 years old, normal-weight and overweight/obese. Every subject participated in two meal-challenge-tests with high-carbohydrate (HC), and normo-carbohydrate (NC) or high-fat (HF) meals. Fasting and postprandial blood was collected for a further 240 min, to determine adiponectin, leptin and total ghrelin concentrations. Results. In normal-weight individuals after HC-meal we observed at 60min higher adiponectin concentrations (12,554 ± 1531 vs. 8691 ± 1070 ng/mL, *p* = 0.01) and significantly (*p* < 0.05) lower total ghrelin concentrations during the first 120 min, than after HF-meal intake. Fasting and postprandial leptin levels were significantly (*p* < 0.05) higher in overweigh/obese men. Leptin concentrations in normal-weight men were higher (2.72 ± 0.8 vs. 1.56 ± 0.4 ng/mL, *p* = 0.01) 180 min after HC-meal than after NC-meal intake. Conclusions. Our results suggest that in normal-body weight men we can expect more beneficial leptin, adiponectin, and total ghrelin response after HC-meal intake, whereas, in overweight/obese men, the HC-meal intake may exacerbate the feeling of hunger, and satiety may be induced more by meals with lower carbohydrate content.

## 1. Introduction

Obesity is a chronic disease resulting from excess fat accumulation. Currently, adipose tissue is recognized as a major endocrine organ and many hormones, growth factors, and cytokines are synthesized and secreted into the circulation by the cells of subcutaneous and visceral adipose tissue. These factors (called adipokines) show auto-, endo-, and paracrine activity and regulate energy homeostasis and insulin sensitivity, inflammatory processes, glucose and lipid metabolism, blood pressure, blood coagulation and proliferation, cell differentiation, as well as apoptosis processes [1,2,3]. The central nervous system receives peripheral signals through numerous receptors, especially from the gastrointestinal tract and adipose tissue, in response to the current energy status and in response to changes in the body energy status [4]. 

Energy balance regulation also has a short-term component and includes metabolic and hormonal changes induced by food consumption. Current studies suggest that ghrelin and leptin and their interactions seem to play a key role in appetite regulation. The increase of leptin hypothalamic expression results in a decrease in ghrelin and adiponectin concentrations [5]. 

The main ghrelin activity is associated with hunger stimulation and stimulation of growth hormone secretion; it also influences energy homeostasis [4]. The highest ghrelin concentrations are observed before a meal intake. During starvation, ghrelin concentrations increase, and they are reduced 60 to 120 min after meal intake [6]. Ghrelin concentrations depend also on the diet energy value, and a postprandial decrease of ghrelin levels is proportional to the meal energy value [7]. Furthermore, ghrelin concentrations depend on the content of essential nutrients in the diet—proteins, fats and carbohydrates—but their influence is not thoroughly known. It seems that after high-carbohydrate and high-protein meals, ghrelin concentrations may be significantly reduced, compared to the high-fat meals [8]. 

Adiponectin synthesis is stimulated by insulin, insulin-like growth factor and peroxisome proliferator-activated receptor gamma (PPARγ-receptor antagonist [9]. Decreased adiponectin synthesis and secretion are observed in both high-energy [10] and high-fat diets [11]. Decreased adiponectin concentrations in peripheral blood are also observed with increased body mass index (BMI), and it increases with body weight reduction [12]. 

Taking into consideration the functions of leptin, ghrelin, and adiponectin, in obese individuals we would expect higher concentrations of ghrelin and adiponectin and lower leptin levels, but the tendencies are reversed [12]. Moreover, in obesity, the metabolic and hormonal response in the postprandial state may differ from the changes observed in subjects with normal body weight [13,14,15,16,17]. The mechanisms of these phenomena are not completely known. People spend most of the day in the postprandial state; therefore, meals that induce the longest possible satiety and an advantageous metabolic response are important in both the prevention and treatment of obesity.

The aim of our study was to evaluate the hormonal changes after meals of varying carbohydrate and fat content, in men with normal body weight and in men who are overweight/obese (in cross-over study design). 

## 2. Materials and Methods

This study is a part of our larger project, which is registered at www.clinicaltrials.gov as NCT03792685, and all methods have been previously described in details [13,14,17,18,19,20,21].

### 2.1. Ethics

The study protocol was approved by the local Ethics Committee (Medical University of Bialystok, Poland, R-I-002/35/2009). All aspects of the study were performed in accordance with the ethical standards set forth in the Declaration of Helsinki of 1975, revised in 2013. Written informed consent was obtained from all participants prior to inclusion in the study.

### 2.2. Study Participants

The study included 46 men, 23 with normal body weight (N) and 23 who were overweight/obese (O/O), ranging in age between 21 and 58 years old. Excluded from the study were any subjects suffering from glucose metabolism disorders, endocrine disorders, liver or renal failure, digestive system diseases, or any other diseases that could influence the study results (including people with history of any gastroenterological and bariatric surgeries) as well as individuals who were receiving pharmaceutical treatment (or any other products with unknown impact on metabolism). Only men were enrolled, since the factors to be analyzed may be characteristic of sexual dimorphism. The study population characteristics are presented in Table 1.

### 2.3. Study Procedures

Based on BMI, the men were divided into two groups (N and O/O). Subsequently, participants were randomly assigned to one of two sub-groups: Group I comprised 11 men with normal body weight (N1) and 12 overweight/obese men (O/O1), while Group II comprised 12 men with normal body weight (N2) and 11 overweight/obese men (O/O2). The crossover method was used to carry out the study. Subjects from Group I received a standardized high-carbohydrate (HC) meal (Nutridrink Fat Free, Nutricia, Poland) and an isocaloric (450 kcal) normo-carbohydrate (NC) meal (Cubitan, Nutricia, Poland). Similarly, men from Group II received the same standardized HC-meal (Nutridrink Fat Free, Nutricia, Poland) and an isocaloric (450 kcal) HF-meal (Calogen, Nutricia, Poland). Subjects received meals in random order, at 1–2 weeks intervals. Subjects were asked to avoid coffee, alcohol, and excessive physical activity at least on the day before each test and to maintain their regular lifestyle throughout the study. The meal contents are presented in Table 2.

Subjects arrived at the laboratory between 8:00 and 8:30 in the morning, after at least 12-h fasting, Each participant’s height and weight measurements and body composition analysis (using the bioimpedance method, InBody 220, Biospace, Korea) were carried out. A peripheral venous catheter was placed in the elbow crook and before receiving the standardized meal, venous blood was collected to determine the fasting adiponectin, leptin, and total ghrelin concentrations. Then the subjects received a randomly assigned meal (at room temperature), with a recommendation to consume it within 10 min. Venous blood was drawn 30, 60, 120, 180, and 240 min after meal consumption to determine postprandial adiponectin, ghrelin, and leptin levels. The blood preparation and laboratory procedures were in accordance with the recommendations of the laboratory kits. The concentrations were determined using the following methods: total adiponectin—radioimmunoassay (Human Adiponectin RIA, Millipore, USA); leptin—immunoenzymatic method (Human Leptin ELISA, BioVendor, Czech Republic); total ghrelin—radioimmunometric method (Ghrelin (total) RIA, Millipore, USA). Biochemical analyses were performed at the Laboratory of the Department of Endocrinology, Diabetology and Internal Medicine, Medical University of Bialystok, Poland.

Statistical analysis. Descriptive statistics, including mean and its standard error (SE), were calculated for all numerical features representing concentrations of interest, which underwent further, consecutive steps of the analysis. The aim of the study was to evaluate whether postprandial hormonal responses differ significantly when the types of meals and patients’ characteristic were used as a grouping factor. We stated two main null hypotheses: (1) the type of meal has no influence on postprandial metabolic response in normal body weight and overweight/obese men (the Is of participants were analyzed separately), (2) there is no statistically significant difference in postprandial hormonal response to a particular meal in normal body weight and overweight/obese men (the meal types were analyzed separately). The first hypothesis was verified for the following pairs of meals: HC vs. NC in Group I, and HC vs. HF in Group II. The procedure was conducted twice—for normal body weight and overweight/obese subjects—and, since both meals were given to the same individuals, the lack of independence was taken into consideration, resulting in the choice of statistical tests. Either one-way ANOVA (analysis of variance) or Wilcoxon signed-rank test (both for paired samples) was carried out, depending on fulfillment of the condition of the normality of the variables’ distribution, analyzed with the Shapiro–Wilk test. The second hypothesis was verified for the investigated meal types: HC, NC, and HF. The goal was to investigate whether there are any statistically significant differences in postprandial hormonal response between normal body weight and overweight/obese men. To test the stated hypothesis we used the one-way ANOVA or Wilcoxon rank-sum test (both for unpaired samples)—dependently on the fulfillment of the condition of normality of the variables’ distribution and the homogeneity of variances. The homogeneity of variances was verified with the Levene test. To address the issue of multiple hypothesis testing, the false discovery rate p-value adjustment method was used [22]. For all calculations, the alpha level was set at 0.05. The areas under the curve (AUCs) were calculated using a trapezoidal method and underwent the same analysis schema, like the rest of the features. 

## 3. Results

The fasting and postprandial differences in adiponectin concentrations between normal body weight and overweight/obese individuals were not significant (Figure 1A,B). However, in subjects with normal body weight, we noted significantly higher (*p* = 0.01) adiponectin concentrations 60 min after the HC-meal than after the HF-meal intake (Figure 1B), while in overweight/obese men, we observed significantly higher (*p* = 0.03) adiponectin levels 120 min after the HF-meal than after the HC-meal intake. 

In normal body weight participants, we observed significantly higher (*p* = 0.01) leptin concentrations 180 min after HC-meal than after NC-meal intake (Figure 2A). In overweight/obese men, although leptin concentrations before the HC-meal were significantly higher (8.44 ± 1.68 vs. 7.07 ± 1.51 ng/mL, *p* = 0.01), postprandially we did not observe any significant differences. The AUC for postprandial leptin levels was significantly higher after the HC-meal intake than after the NC-meal (670 ± 220 vs. 391 ± 103, *p* = 0.04) in the N group. When we compared the postprandial leptin levels between the HC and HF-meals, we found that men with normal body weight showed a tendency, which was on the margin of significance (*p* = 0.05), to higher leptin concentrations 240 min after the HC-meal intake (Figure 2B). In overweight/obese subjects, we did not observe any significant differences in postprandial leptin concentrations dependent on meal type. Leptin levels in O/O men were significantly higher than in N subjects, at fasting state and during the further 4 h of all of the meal challenge tests (Figure 2A,B).

The total ghrelin concentration analysis in Group I showed that there were not any significant differences dependent on meal type in N and O/O men (Figure 3A). However, in Group II, we found that in N subjects the total ghrelin concentrations were significantly lower after the HC-meal intake than after the HF-meal intake (Figure 3B). Lower values were observed at fasting state (744 ± 79 vs. 884 ± 105 ng/mL; *p* = 0.02) and during the first 120 min of the test (30 min: 701 ± 56 vs. 929 ± 101 ng/mL, *p* = 0.0005; 60 min: 637 ± 57 vs. 787 ± 82 ng/mL, *p* = 0.0005; 120 min: 673 ± 64 vs. 804 ± 93 ng/mL, *p* = 0.03). At 240 min the total ghrelin levels were significantly higher after the HC-meal intake than after the HF-meal (860 ± 92 vs. 748 ± 79 ng/mL, *p* = 0.03). In addition, the AUC for postprandial ghrelin levels was significantly lower after the HC-meal intake than after the HF-meal (174,263 ± 15,962 vs. 202,764 ± 24,214, *p* = 0.007) in N men. In O/O individuals we did not find any significant differences between total ghrelin concentrations after the HC-meal and the HF-meal intake, except at 240 min of the test, when total ghrelin concentrations were significantly lower after the HF-meal consumption (774 ± 77 vs. 586 ± 52 ng/mL, *p* = 0.003) (Figure 3B). In Group I, we did not notice any differences in total ghrelin concentrations between N men and O/O men (Figure 3A). In Group II (Figure 3B) 30 min after the HF-meal intake 30 we observed lower total ghrelin levels in O/O men than in N individuals.

## 4. Discussion

The conducted experiment revealed the differences in postprandial adiponectin, leptin and total ghrelin response dependently on the macronutrients meal composition, and also dependently on the body weight. In normal-weight individuals after an HC-meal, we observed higher adiponectin and lower total ghrelin concentrations, than after an HF-meal intake. After the HC-meal intake, we noted also higher leptin concentrations than after NC-meal intake, in normal body weight men. However, higher fasting and postprandial leptin levels we observed in overweight/obese individuals. Investigated hormones and adipokines are involved in energy homeostasis regulation, and play a crucial role in body fat accumulation. 

Pathological amounts of adipose tissue lead to cardiovascular diseases, lipid disorders, and type 2 diabetes, which are significant medical problems [3,23,24,25]. Due to the dynamic nature of obesity [26], it is necessary to broaden our knowledge about the physiological mechanisms involved in energy balance regulation. Hunger and satiety are regulated by the central nervous system’s receipt of central and peripheral signals [27,28,29], which are influenced by environmental factors, including diet [29,30]. In our study, we have observed that the postprandial levels of investigated factors depend on the meal content and may differ in N and O/O men. In N men we noted higher adiponectin levels after the HC-meal intake than after the HF-meal; while in O/O subjects, adiponectin concentrations were significantly higher after the HF-meal intake than after the HC-meal. Our results contrast with those of some other studies, which showed that serum levels of adiponectin are very stable and are not acutely affected by oral glucose or fat load [31,32], but these differences between findings may result from the different nutritional composition of the standardized meals. 

We did not observe any significant differences in fasting adiponectin concentrations between N subjects and O/O subjects, although it is generally accepted that people with obesity are characterized by lower adiponectin concentrations [33,34]. However, it was also shown that adiponectin levels in obese and metabolically healthy individuals are comparable with adiponectin concentrations in normal body weight individuals [35], what may explain the lack of significant differences in fasting adiponectin concentrations in our observations. 

Another hormone secreted primarily by adipose tissue that is involved in the regulation of body energy homeostasis is leptin. In O/O men, we noted significantly higher leptin concentrations at fasting state and throughout the meal challenge tests. The higher levels in the postprandial period in O/O men were undoubtedly the result of higher baseline values, which may be a consequence of positive energy balance and leptin resistance development [36]. In the context of hunger and satiety regulation, more important are the postprandial changes in leptin levels. In Group I, we have noted significantly higher postprandial leptin concentrations only after the HC-meal in N subjects. We did not observe this effect in O/O men, even if the baseline leptin levels before the HC-meal intake were significantly higher. Our results are inconsistent with the results of Marzullo et al. [15], who showed a slight increase in leptin concentrations in subjects with obesity for 2 h after the HC-meal consumption, whereas, in individuals with normal body weight, authors noted lower leptin concentrations than their baseline values. However, other researchers [16] noticed a greater increase in leptin concentrations after the HC-meal in women with normal body weight, compared to women who were obese, in whom leptin concentrations started to increase just 4 h after HC-meal intake, which is in line with our results. It is worth emphasizing that, in the cited study, the authors considered the HC-meal a meal in which carbohydrates covered 53% of the meal energy, which corresponds better to our NC-meal composition. 

After the HF-meal the observations from our study differ from the results obtained by some other researchers, who observed an increase in leptin concentrations after the HF-meal in subjects with normal body mass, whereas in obese individuals they noted a significant decrease in leptin concentrations [37]. However, our results seem to be comparable to the results of studies conducted by Marzullo et al. [15], who showed that postprandial leptin levels in subjects with normal body weight, after HF-meal intake, remained unchanged for 2 h into the test, while in obese individuals postprandial leptin concentrations decreased. The other authors showed that, after a mixed meal intake, leptin concentrations in people with normal body weight were reduced, and an increase was noted from 2 to 8 h after the meal intake [38]. Kim S. et al. [39] noted reduced leptin levels in women after a meal in which carbohydrates accounted for 60% of the energy. Other authors [40] have demonstrated that in obese individuals leptin levels are reduced for the first 2 h after a mixed meal intake, and it returns to baseline values after the next 6 to 12 h. In men with normal body weight, we have noted an increase in postprandial leptin levels only after meals that contained carbohydrates (the difference 240 min after meal intake was on the margin of significance), and this observation seems to be consistent with the results of Monteleone et al. [41] and Romon et al. [42], who showed that in people with normal body weight and BMI ≤ 27 kg/m^2^, leptin concentrations after an HC-meal were higher than after consumption of the HF-meals, while leptin levels decreased. Other researchers have shown that after a high-fat meal intake, leptin concentrations were reduced for the first 2 h but then a significant increase was noticed, with a maximum concentration 8 h after meal intake [43]. A study by Raben et al. [44] showed that the decrease in leptin levels is more pronounced after the HC-meal than after the HF-meal, whereas a greater increase in leptin concentrations in relation to fasting values was observed only at 195 min after the HC-meal intake. 

Taken together, the results of these studies are inconsistent and it seems that comparisons of leptin concentrations between studies make sense only with similar study protocols and similar nutrient contents of tested meals, but also with comparable study groups, since leptin concentrations are also determined by sexual dimorphism [45]. 

Besides leptin, which shows anorexigenic activity (decreasing appetite) [46], also ghrelin plays an important role, and both hormones together participate in the regulation of hunger and satiety [47]. Ghrelin is a gastrointestinal hormone with a well-documented orexigenic effect [48]. In our study, despite the apparently higher mean fasting total ghrelin concentrations in subjects with normal body weight than in overweight/obese men, these differences were not statistically significant, probably due to a too-small study sample, which was a major limitation of our study. Other researchers [49] have shown that people with normal body weight tend to have higher ghrelin concentrations. Moreover, in our study we did not notice any important differences in total ghrelin levels between subjects with normal body weight and overweight/obese subjects, except one time-point, which was 30 min after HF-meal intake, when in the overweight/obese men the total ghrelin level decreased, while surprisingly in the normal body weight men it increased. Our results are in line with the study by Heinonen M. et al. [50], who in subjects with obesity and metabolic syndrome, did not observe any decrease in ghrelin levels after HC-meal intake, compared to subjects with normal body weight. An experiment conducted by Zwirska-Korczala et al. [49] showed a more pronounced decrease in ghrelin concentrations after mixed meals in normal body weight subjects than in obese participants, but the study group consisted exclusively of women and different changes depending on sex cannot be excluded. Moreover, the investigated meals differed in essential nutrient content from the meals used in our study, and the test lasted for 120 min. 

When we compared the HC-meals with the HF-meals, we found that we could expect a more beneficial response in lean subjects after the HC-meal intake, while after the HF-meal the total ghrelin levels tended to be even higher than at fasting state, and started to decrease at 60 min of the test. The differences were statistically significant also at baseline before meal intake, in the same study group, probably due to the daily variations of total ghrelin levels. Importantly, in overweight/obese men from Group II, we have noted that ghrelin levels decreased postprandially after both meals, but the decrease was more pronounced after the HF-meal intake than after the HC-meal consumption. These results differ from those obtained by Marzullo et al. [15], who showed that the decrease in ghrelin levels after an HF-meal is more pronounced in subjects with normal body weight than in subjects with obesity. It seems that the difference in results can be caused by different compositions of the tested meals, which contained a lower percentage of fat than meals in our study.

The major limitations of our study are the small sample size and the fact that we could not create one study group, in which all volunteers would receive all of the three investigated meals. The main reason is that the presented study is a part of our larger project, with very long and laborious protocol procedures, what limited the final number of volunteers who would agree to participate in the all of the meal challenge tests, with various meals intake. Therefore, it was needed to divide participants into two groups (Group I and Group II), if we aimed to compare the postprandial responses to different meals in the same individuals, following a crossover study design. The other limitations include enrolling only the male participants and the liquid form of meals. These limitations were actually intended and allowed us to reduce the impact of possible confounding factors, such as the influences of sex hormones and sex differences; or to decrease the time of meal digestion and absorption, to not discourage volunteers with a long time spent at each visit etc. However, limitations mentioned above could affect our results, and therefore, our observations regarding the differences in personal postprandial hormonal response dependently on BMI and meal content need further investigation.

## 5. Conclusions

In conclusion, our study showed that postprandial concentrations and/or changes in concentrations of adiponectin, leptin, and total ghrelin may differ depending on current body energy status, as well as on meal macronutrients content. Our findings suggest that in men with normal body weight we can expect a more beneficial hormonal response after an HC-meal intake, whereas in overweight/obese men, more beneficial effects we have observed after meals with lower carbohydrate and higher fat content. Thus, the practical implications of our study may be the recommendation for overweight/obese people to limit the consumption of high-carbohydrate meals, in exchange for meals in which less than 50% of the energy value comes from carbohydrates.

## Figures and Tables

**Figure 1 nutrients-11-00493-f001:**
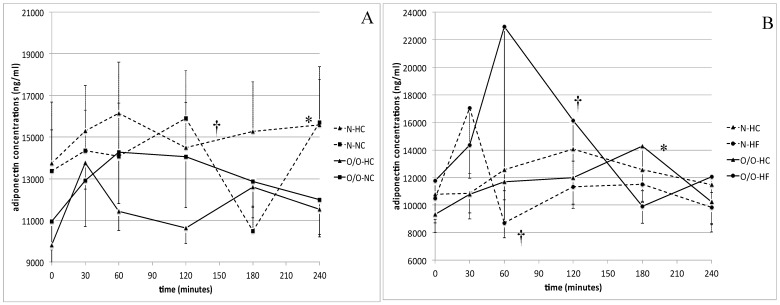
(**A**) Adiponectin concentrations (ng/mL) in men with normal body weight (N, the broken line) and overweight/obese people (O/O, the solid line) in fasting state (time 0 min) and after consumption (time 30–240 min) of a high carbohydrate meal (HC) and a normal carbohydrate meal (NC). The results are presented as mean values ± SE. * The comparison between study groups N and O/O, *p* < 0.05. † The comparison between meals HC and NC, *p* < 0.05. (**B**) Adiponectin concentrations (ng/mL) in men with normal body weight (N, the broken line) and overweight/obese people (O/O, the solid line) in fasting state (time 0 min) and after consumption (time 30–240 min) of a high carbohydrate meal (HC) and a high fat meal (HF). The results are presented as mean values ± SE. * The comparison between study groups N and O/O, *p* < 0.05. † The comparison between meals HC and HF, *p* < 0.05.

**Figure 2 nutrients-11-00493-f002:**
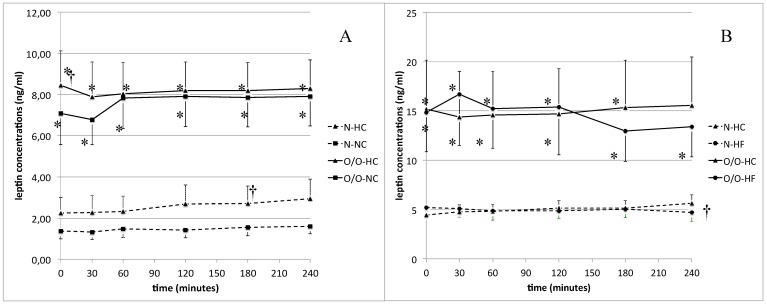
(**A**) Leptin concentrations (ng/mL) in men with normal body weight (N, the broken line) and overweight/obese people (O/O, the solid line) in fasting state (time 0 min) and after consumption (time 30–240 min) of a high carbohydrate meal (HC) and a normal carbohydrate meal (NC). The results are presented as mean values ± SE. * The comparison between study groups N and O/O, *p* < 0.05. † The comparison between meals HC and NC, *p* < 0.05. (**B**) Leptin concentrations (ng/mL) in men with normal body weight (N, the broken line) and overweight/obese people (O/O, the solid line) in fasting state (time 0 min) and after consumption (time 30–240 min) of a high carbohydrate meal (HC) and a high fat meal (HF). The results are presented as mean values ± SE. * The comparison between study groups N and O/O **p* < 0.05. † The comparison between meals HC and HF *p* < 0.05.

**Figure 3 nutrients-11-00493-f003:**
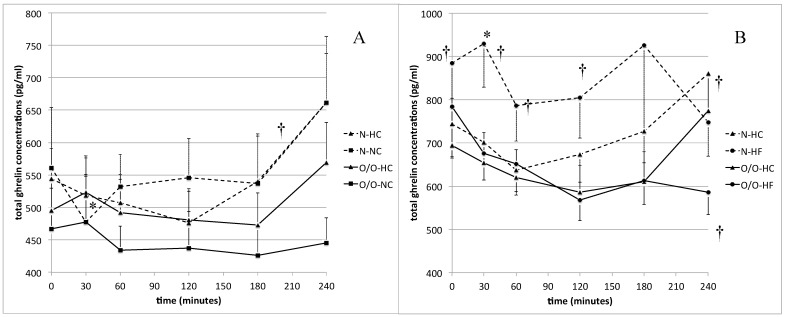
(**A**) Total ghrelin concentrations (ng/mL) in men with normal body weight (N, the broken line) and overweight/obese people (O/O, the solid line) in fasting state (time 0 min) and after consumption (time 30–240 min) of a high carbohydrate meal (HC) and a normal carbohydrate meal (NC). The results are presented as mean values ± SE. * The comparison between study groups N and O/O, *p* < 0.05. † The comparison between meals HC and NC, *p* < 0.05. (**B**). Total ghrelin concentrations (ng/mL) in men with normal body weight (N, the broken line) and overweight/obese people (O/O, the solid line) in fasting state (time 0 min) and after consumption (time 30–240 min) of a high carbohydrate meal (HC) and a high fat meal (HF). The results are presented as mean values ± SE. * The comparison between study groups N and O/O, *p* < 0.05. † The comparison between meals HC and HF, *p* < 0.05.

**Table 1 nutrients-11-00493-t001:** The study population characteristic.

		Normal-weight Men	Overweight/Obese Men	*p*-Value
Group I	*n*	11	12	
	Age (years)	33 ± 2	40 ± 2	0.01
	BMI	23.8 ± 0.5	31.4 ± 1.5	0.0002
	Body fat content (%)	17.9 ± 1.0	28.6 ± 1.7	0.00003
Group II	*n*	12	11	
	Age (years)	33 ± 3	36 ± 3	0.24
	BMI	23.9 ± 0.2	33.7 ± 2.2	0.000001
	Body fat content (%)	18.6 ± 1.5	31.9 ± 2.7	0.0002

The results are presented as mean values ± SE.

**Table 2 nutrients-11-00493-t002:** The energy and macronutrients composition of meals.

	High-carbohydrate Meal	Normo-carbohydrate Meal	High-fat Meal
Energy (kcal)	450	450	450
Carbohydrate (g)	100.5	51.1	4.0
Carbohydrate (% of total energy)	89.3	45.1	4.0
Fat (g)	0	12.6	47.5
Fat (% of total energy)	0	25.2	96
Protein (g)	12	36	0
Protein (% of total energy)	10.7	29.7	0
Fiber (g)	0	0.1	0

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
