# Peer review of "The Differences in Postprandial Serum Concentrations of Peptides That Regulate Satiety/Hunger and Metabolism after Various Meal Intake, in Men with Normal vs. Excessive BMI"

_nutrients, 2019, doi:10.3390/nu11030493_

Round 1

Reviewer 1 Report

Overall the manuscript is well presented, but at some instances, clarity can be enhanced.

I suggest to use different symbol to describe statistical differences when compare different groups.

Line 22 in the Abstract section please corrects the sentence “The study crossover study”.

Author Response

We wish to thank the Editors and Reviewers for their valuable contributions and comments. Our manuscript has been significantly improved by incorporating their suggestions.

We have submitted a revised version that addresses the comments made by the Reviewers and Editors. Our responses to each comment are detailed below.

Reviewer 1

Overall the manuscript is well presented, but at some instances, clarity can be enhanced.

Response:  The manuscript has been clarified and improved, especially the Results and Discussion sections.

I suggest to use different symbol to describe statistical differences when compare different groups.

Response:  We used different symbols to describe statistical differences, as suggested (please see Figures and lines 344, 349, 410, 415, 489, 495)

Line 22 in the Abstract section please corrects the sentence “The study crossover study”.

Response:  The mistake has been corrected (please see line 22).

Reviewer 2 Report

I found the manuscript by Adamska-Patruno et al. interesting, and was grateful for the opportunity to review it. The overall premise of comparing postprandial peptide responses of meals with different macronutrient distributions in individuals that differ my BMI very intriguing. Further, it does appear that noteworthy group- or meal-based difference may exist in these markers in the postprandial period. However, I do have some concerns regarding this manuscript.

Major concerns:

I do not understand the rationale behind the current study design. At some level, it is a randomized cross-over study, but in reality it is not. While participants are randomized into either Group 1 or 2, ALL participants receive the HC (essentially control) meal first, and then receive one of the intervention meals (either NC or HF). So, the order between control and one of the intervention meals was not randomized or even varied, which is a weakness, since there could be some sort of time/order/carry-over effect between trials that is not controlled for via randomization. Additionally, not all participants consume all meals, thereby introducing variability and lowering statistical power. I think this study would be much stronger if it had simply been designed as a three-trial randomized cross-over study in which all participants completed the HC, NC, and HF meals in random order. Then we would not have to worry about potential confounding group differences or time/order effects.

If I am understanding correctly, the authors utilized a t-test (or non-parametric equivalent) for all of the statistical analyses. I think that is not a sufficient statistical approach to assess this complex dataset, and very likely makes the results vulnerable to Type 1 error, since so many comparisons were run. It seems to me that there are potential time-, meal-, and group-based effects. This calls for at least some form of ANOVA, if not something more complex. Specifically, it is not appropriate, in my opinion, to “zoom in” to look at individual time-point differences without first confirming that there is a significant interaction between factors or at least some type of main effect.

Although I think the results need to be re-analyzed with a more complex statistical test anyway, I also have concerns with the statistical interpretations in their current form. Numerous times the authors refer to a “tendency toward difference”. If the alpha level is truly set at 0.05, as the authors state, then it is not appropriate to say that p values greater than this reflect a difference, even a marginal difference, tendency or trend. The authors even go so far to say that “…the trends observed in our study confirm that…”. This interpretation is too strong. This issue with liberal interpretation, in combination with the risk of type 1 error associated with the many t-tests, makes me wonder if many of the key results observed occurred simply by chance.

Minor points:

I think actual results (Mean ± SD, p values, etc.) should be included in the Results of the Abstract.

The first paragraph of the Intro is pretty long. It could probably divided into multiple small paragraphs.

Line 68-69: This statement seems very subjective.

The statement on Lines 70-71 is critical to the context of the study, and deserves to be elaborated on and specific studies cited.

Did the authors make hypotheses before the study began? These merit inclusion.

Was an a priori sample size estimation conducted? How did you arrive at n = 46, and n = 23 in each group? Can we be sure this was sufficient to detect differences between groups?

Are BMI and Body fat content different between the 2 obese groups? They look like they could be.

What was the rationale behind the test meals utilized? What were the kcal and macros based on? Previous studies? The kcal seems somewhat low compared to other postprandial studies. Did the authors consider normalizing the meal to body weight? I think this could be very relevant since groups are innately different in body size/mass.

Did the authors make any efforts to control for diet and PA between groups and leading up to meal trials? How long was the fast?

Was blood collected with an IV catheter or repeated venipunctures?

Thank you for testing for the normality of your data and adjusting accordingly.

In Figure 1B, the error bar for the O/O-HF group is very large. Have you considered an objective analysis to determine whether it is being influenced by an outlier?

I recommend beginning the Discussion with a clear summary of the main findings.

I recommend dividing some of the larger paragraphs in the Discussion into multiple smaller paragraphs.

Please include a Limitations section.

Author Response

We wish to thank the Editors and Reviewers for their valuable contributions and comments. Our manuscript has been significantly improved by incorporating their suggestions.

We have submitted a revised version that addresses the comments made by the Reviewers and Editors. Our responses to each comment are detailed below.

Reviewer 2

I found the manuscript by Adamska-Patruno et al. interesting, and was grateful for the opportunity to review it. The overall premise of comparing postprandial peptide responses of meals with different macronutrient distributions in individuals that differ my BMI very intriguing. Further, it does appear that noteworthy group- or meal-based difference may exist in these markers in the postprandial period. However, I do have some concerns regarding this manuscript.

Response:  We thank the Reviewer for these comments.

Major concerns:

I do not understand the rationale behind the current study design. At some level, it is a randomized cross-over study, but in reality it is not. While participants are randomized into either Group 1 or 2, ALL participants receive the HC (essentially control) meal first, and then receive one of the intervention meals (either NC or HF). So, the order between control and one of the intervention meals was not randomized or even varied, which is a weakness, since there could be some sort of time/order/carry-over effect between trials that is not controlled for via randomization. Additionally, not all participants consume all meals, thereby introducing variability and lowering statistical power. I think this study would be much stronger if it had simply been designed as a three-trial randomized cross-over study in which all participants completed the HC, NC, and HF meals in random order. Then we would not have to worry about potential confounding group differences or time/order effects.

Response: The study design has been described in details in our previously publish articles, that we have referenced to, with the same study group. We agree with Reviewer that it could be confusing, therefore the description of study design has been clarified. It was a translation error and we have clarified that subjects received meals in random order, at 1-2 weeks intervals (please see line 168), what was also mentioned in our previously published and referenced articles.

We strongly agree with Reviewer that the best would be to investigate a group that would participate in all of the 3 meals-challenge-tests, but results that we present in this manuscript are a part of our larger project, with long and laborious study procedures. Therefore it is very difficult to find volunteers who would agree to participate in all of the meal-challenges visits, what has been mentioned as a major limitation of our study (please see lines 742-748). We had to divide participants into the two study groups, with two different meals intake, if we wanted to compare the postprandial responses to different meals in the same individuals, following a crossover study design.

If I am understanding correctly, the authors utilized a t-test (or non-parametric equivalent) for all of the statistical analyses. I think that is not a sufficient statistical approach to assess this complex dataset, and very likely makes the results vulnerable to Type 1 error, since so many comparisons were run. It seems to me that there are potential time-, meal-, and group-based effects. This calls for at least some form of ANOVA, if not something more complex. Specifically, it is not appropriate, in my opinion, to “zoom in” to look at individual time-point differences without first confirming that there is a significant interaction between factors or at least some type of main effect.

Response:  It was our mistake and methods were not properly described. In previous version of manuscript we just referenced to our other articles with detailed descriptions, but in revised version we have corrected and provided more details about statistical analyses (please see lines 209-233). Moreover, please find below an answer from one of the author- a statistician, who was responsible for the data analysis.

Treating the presented results as a preface to broader research made the authors conduct the statistical analysis in the matter of analyzing particular time-points – making the studied groups bigger is their goal and that’s when the analysis taking into consideration factors – such as “group” and “age” (potential confounding factor) will take place.

The authors main interest (when concerning hypotheses testing) was not to commit type I errors and that is why p-value correction was used. False Discovery Rate p-value correction procedure, which was applied, is considered to be powerful. Nevertheless, it reduces power. Summarizing, the authors controlled for type I errors at the expense of type II errors (power).

Although I think the results need to be re-analyzed with a more complex statistical test anyway, I also have concerns with the statistical interpretations in their current form. Numerous times the authors refer to a “tendency toward difference”. If the alpha level is truly set at 0.05, as the authors state, then it is not appropriate to say that p values greater than this reflect a difference, even a marginal difference, tendency or trend. The authors even go so far to say that “…the trends observed in our study confirm that…”. This interpretation is too strong. This issue with liberal interpretation, in combination with the risk of type 1 error associated with the many t-tests, makes me wonder if many of the key results observed occurred simply by chance.

Response: We agree with Reviewer and we have corrected the interpretation of our results throughout the manuscript. We have critically revised our paper, and interpretations regarding any trends and tendencies, which were not statistically significant, have been removed. Only one difference, which was on the margin of significance (p=0.05), has been mentioned (please see lines 355-358, 611-617). We revised also the Conclusions section, and “…the trends observed in our study confirm that…” we have replaced with “Our findings suggest…” (please see line 798).

Minor points:

I think actual results (Mean ± SD, p values, etc.) should be included in the Results of the Abstract.

Response:  The mean ± SD and p values have been included in abstract (please see lines 27-32).

The first paragraph of the Intro is pretty long. It could probably divided into multiple small paragraphs.

Response: The first paragraph has been divided as suggested, please see the Introduction section.

Line 68-69: This statement seems very subjective.

Response:  The sentence has been reconstructed, to be more objective (please see lines 114-116).

The statement on Lines 70-71 is critical to the context of the study, and deserves to be elaborated on and specific studies cited.

Response:  We cited some studies as suggested (please see line 118), but because results are very often inconsistent, we just mentioned that postprandial response may differ between normal body weight and obese individuals, what was also one of the aim of our investigation, and to not extend the Introduction section we discussed it widely in the Discussion section.

Did the authors make hypotheses before the study began? These merit inclusion.

Response:  The stated hypotheses have been included in the statistical analysis description (please see lines 213-217).

Was an a priori sample size estimation conducted? How did you arrive at n = 46, and n = 23 in each group? Can we be sure this was sufficient to detect differences between groups?

Response:  Addressing the issue of sample size, the presented results can be treated as a preliminary study and it is planned to develop the matter. Furthermore, the employment of the FDR p-value correction procedure, prevented the authors from committing type I errors.

Are BMI and Body fat content different between the 2 obese groups? They look like they could be.

Response:  The differences between Group I and Group II were not estimated, since it was not the aim of our study, and in our opinion it was not needed to compare groups. The aim of study was to evaluate within each group the differences between normal-body-weight and overweight/obese men, and between two investigated meals.

What was the rationale behind the test meals utilized? What were the kcal and macros based on? Previous studies? The kcal seems somewhat low compared to other postprandial studies. Did the authors consider normalizing the meal to body weight? I think this could be very relevant since groups are innately different in body size/mass.

Response:  We agree with Reviewer that there are many different possibilities regarding study designs, with different energy and macronutrients meal compositions and calculations. Some Authors investigated meals with higher, some with lower kcal content etc. We chose 450 kcal meals as a similar to the regular real-life breakfasts. We decided to investigate isocaloric meals, without normalizing to the body weight, because it was our aim to compare responses between normal body weight and overweight/obese men to exactly the same meals intake, what is also very common in studies comparing lean and obese subjects.

Did the authors make any efforts to control for diet and PA between groups and leading up to meal trials? How long was the fast?

Response:  Subjects were asked to stay for at least 12-h fasting before every meal test visit, to avoid coffee, alcohol, and excessive physical activity at least on the day before each test and to maintain their regular lifestyle throughout the study. It was recorded in the subject’s diaries and controlled before every meal challenge test visit. This information has been updated in our manuscript (please see lines 168-170).

Was blood collected with an IV catheter or repeated venipunctures?

Response:  The blood was collected with a peripheral venous catheter, which was placed in the elbow crook. This information has been updated in text (please see lines 177-178).

Thank you for testing for the normality of your data and adjusting accordingly.

Response:  We thank Reviewer for this comment.

In Figure 1B, the error bar for the O/O-HF group is very large. Have you considered an objective analysis to determine whether it is being influenced by an outlier?

Response:  Following the definition of the standard error it is highly probable, that evidently highest mean value at point 60’ (O/O-HF) is followed by the highest value of standard deviation. Therefore, the standard error reaches higher levels as it is presented in Fig 1B.

I recommend beginning the Discussion with a clear summary of the main findings.

Response:  We reconstructed the beginning of the Discussion, with a summary of our main findings, as suggested (please see lines 498-505).

I recommend dividing some of the larger paragraphs in the Discussion into multiple smaller paragraphs.

Response:  The Discussion section has been divided for the shorter paragraphs, as recommended (please see the Discussion section).

Please include a Limitations section.

Response:  The limitations have been added (please see lines 742-755).